# Ammonia Cycling and Emerging Secondary Aerosols from Arable Agriculture: A European and Irish Perspective

Vivien Pohl [1,*], Alan Gilmer [1,2], Stig Hellebust [3], Eugene McGovern [2], John Cassidy [4], Vivienne Byers [1], Eoin J. McGillicuddy [1,4], Finnian Neeson [5] and David J. O'Connor [5]

1. Environmental Sustainability and Health Institute, Technological University Dublin, Grangegorman Lower, D07 EWV4 Dublin, Ireland
2. College of Engineering and Built Environment, Technological University Dublin, Bolton Street, D01 K822 Dublin, Ireland
3. School of Chemistry, University College Cork, T12 CY82 Cork, Ireland
4. School of Chemical and BioPharmaceutical Sciences, Technological University Dublin, Grangegorman Lower, D07 H6K8 Dublin, Ireland
5. School of Chemical Sciences, Dublin City University, Glasnevin, D09 E432 Dublin, Ireland
* Correspondence: vivien.pohl@tudublin.ie

**Abstract:** Ammonia ($NH_3$) is a naturally occurring, highly reactive and soluble alkaline trace gas, originating from both natural and anthropogenic sources. It is present throughout the biosphere, yet plays a complicated role in atmospheric acid–base reactions resulting in the formation of inorganic secondary inorganic aerosols (SIAs). While the general mechanisms are recognised, factors controlling the reactions leading to SIA formation are less explored. This review summarises the current knowledge of $NH_3$ sources, emission and deposition processes and atmospheric reactions leading to the formation of SIA. Brief summaries of $NH_3$ and SIA long-range transport and trans-boundary pollution, a discussion of precursor species to SIAs (other than $NH_3$), abiotic and biotic controls and state-of-the-art methods of measurement and modelling of pollutants are also included. In Ireland, $NH_3$ concentrations remained below National and European Union limits, until 2016 when a rise in emissions was seen due to agricultural expansion. However, due to a lack of continuous monitoring, source and receptor relationships are difficult to establish, including the appointment of precursor gases and aerosols to source regions and industries. Additionally, the lack of continuous monitoring leads to over- and underestimations of precursor gases present, resulting in inaccuracies of the estimated importance of $NH_3$ as a precursor gas for SIA. These gaps in data can hinder the accuracy and precision of forecasting models. Deposition measurements and the modelling of $NH_3$ present another challenge. Direct source measurements are required for the parameterization of bi-directional fluxes; however, high-quality data inputs can be limited by local micrometeorological conditions, or the types of instrumentation used. Long-term measurements remain challenging for both aerosols and precursor gases over larger areas or arduous terrains.

**Keywords:** ammonia; nitrogen; aerosols; atmospheric chemistry; pollution; deposition; emissions

## 1. Introduction

Nitrogen ($N_2$) is a vital element of life on Earth, constituting approximately 79% of the atmosphere. It can be found in major and minor pools throughout the biosphere in various forms. However, excessive anthropogenic contributions of various nitrogen (N) compounds have made it one of the four primary pollutants resulting in significant damage to both environmental and human health [1,2]. The other three main classes are sulphur compounds, volatile organic compounds (VOCs) and heavy metals [3]. Species of N can be present as gases and aerosols (solid or liquid) in the atmosphere or alternatively as part of water vapor [2,3]. Species include oxides (nitrogen dioxide ($NO_2$) and nitric oxide (NO) collectively known as $NO_x$), nitric acid ($HNO_3$), nitrate ($NO_3^-$), ammonia ($NH_3$) and

ammonium ($NH_4^+$) [4–7], as well as the highly reactive nitrate radicals ($NO_3$) Among all the forms of N present in the atmosphere, $NH_3$ plays a key role in atmospheric reactions resulting in the formation of secondary inorganic aerosols (SIAs) [8,9].

In the atmosphere, $NH_3$ is a key alkaline constituent, which readily reacts with acidic species present forming ammonium salts, such as ammonium sulphate (($NH_4$)$_2SO_4$), ammonium bisulphate ($NH_4HSO_4$), ammonium nitrate ($NH_4NO_3$) and ammonium chloride ($NH_4Cl$), collectively known as SIAs (secondary inorganic aerosols) [10–13]. SIAs are known to possess the ability to harm both human and environmental health, generating interest from the research community in recent years, especially in the area of air quality and models based around pollutant emissions and effects [14–16]. SIAs can remain in the atmosphere for several weeks, causing atmospheric haze (however, due to Ireland's wet climate, haze formation is not as severe an issue compared with countries with drier climates) [17]. Due to SIA persistence in the atmosphere, it can be transported to much greater distances than $NH_3$ gas.

This paper reviews the current knowledge base and state-of-the-art methodologies in use for both $NH_3$ and SIA detection and modelling. Additionally, it is proposed that in order to design a model capable of accurately predicting SIA concentrations in the atmosphere, the precursor species' dynamics should be included in the model construction. Current models in use require a set of basic parameters to be established, similar to those set forth by the US Environmental Protection Agency in order to obtain a structured, transparent approach, regulating models for specific uses [18]. This would harmonize current models and enhance accuracy and precision of forecasts of pollution events on both local and continental scales within Europe.

## 2. Precursor Species' Dynamics and SIA Formation

### 2.1. Source Appointment and Emission

While a concentrated EU-wide attempt is being carried out for the reduction in N emissions through various legislative measures, such as the Gothenburg Protocol and the National Emission Ceilings (NEC) regulations, there has been little progress in controlling $NH_3$ emissions resulting in increasing air pollution arising from sources such as agriculture within the European Union [12,13]. In Ireland, the Environmental Protection Agency (EPA) currently monitors atmospheric particulate $NH_4^+$ at three sites (Carnsore, County Wexford; Oak Park, County Carlow and Malin, County Donegal) in agreement with the European Monitoring and Evaluation Program (EMEP); however, there is no continuous monitoring network in place for ambient atmospheric $NH_3$ gas concentrations at present [19]. Currently, $NH_3$ is one of few pollutants not covered by the CAFE Directive under ambient air quality and does not fall under the National Ambient Air Quality Network, which is managed by the EPA [20]. This presents major difficulties in mapping $NH_3$ concentrations both on localised and national scales.

Ireland has seen limited work undertaken with regard to $NH_3$ concentrations; however, annual average concentration of $NH_3$ were measured during a study by Kluizenaar et al. (2000) across 40 sites in Ireland and found to be 1.45 $\mu g/m^3$ in 1999 [21]. Annual averages obtained for each site ranged between 0.14 and 7.24 $\mu g/m^3$. From 2013 to 2014, another study performed Doyle et al. (2017) found the annual average concentration of $NH_3$ to be 1.72 $\mu g/m^3$ for 25 study sites in Ireland from June 2013 to July 2014 [19]. The minimum detectable concentration was 0.20 $\mu g/m^3$ and the maximum concentration detected during the study was 10.51 $\mu g/m^3$ over the study period. Measured concentrations from these studies demonstrate a strong correlation between regions of high $NH_3$ concentrations and $NH_3$ hotspots. Additionally, a general increase in average concentrations is also demonstrated both in minimum and maximum concentrations observed.

Among all N species emitted to the atmosphere, $NH_3$ emissions arise from both natural and anthropogenic sources alike. Between all anthropogenic sources, the greatest contributor to atmospheric $NH_3$ is agriculture, accounting for more than 90% of all emissions [10]. Other sources include sewage, biomass burning, fossil fuel combustion and

catalytic converters used in cars [22–24]. Natural emissions of $NH_3$ include sources such as the oceans, forest fires and vegetation. Sources of $NH_3$ can be found throughout the three major reservoirs on Earth: atmosphere, soils/groundwater, and biomass. The terrestrial N cycle consists of soil, flora and fauna pools containing low quantities of N in various forms (in comparison to atmospheric and lithospheric reservoirs) yet still exerts a substantial impact on the natural dynamics of the biogeochemical N cycle [25]. This can result in N being a limiting agent in nutrient uptake.

Generally, plants and vegetation acquire N from the soil in much greater quantity than any other element; however, most plants are only able to utilise N in two of its solid forms: $NH_4^+$ and nitrate ($NO_3^-$) [26,27]. The bioavailability of these inorganic forms of N and the natural dynamics within the cycle can be broken down into six main reactions: mineralisation and immobilisation, nitrification and denitrification, microbial N fixation and volatilization [28]. These reactions can be severely affected by anthropogenic additions of bioavailable N, such as $NH_3$, $NH_4^+$ and $NO_3^-$.

$NH_3$ losses from arable agricultural systems primarily occur through volatilization after the application of organic (manure, slurry and/or urea) and synthetic fertilizers. Other emissions from agricultural systems include emissions related to animal husbandry, such as storage of manure for example. [29,30]. Of the N applied to land, more than 40% of the loss is recorded as $NH_3$ and under specific environmental and edaphic conditions, an average of 10–14% is reported as lost through volatilization from synthetic fertilizer application [2,31–33]. Presently, approximately 100 million metric tonnes of N-based fertilizer is produced per annum globally, in comparison to 1 million metric tonnes 40 years ago [2,34]. Excessive $NH_3$ emissions from anthropogenic sources such as agriculture can lead to biodiversity loss, eutrophication, air pollution and acidification of aquatic and terrestrial environments; and unbalancing N loads throughout the cycle [35–38]. An abbreviated N cycle indicating the role of $NH_3$ in agricultural settings is given in Figure 1.

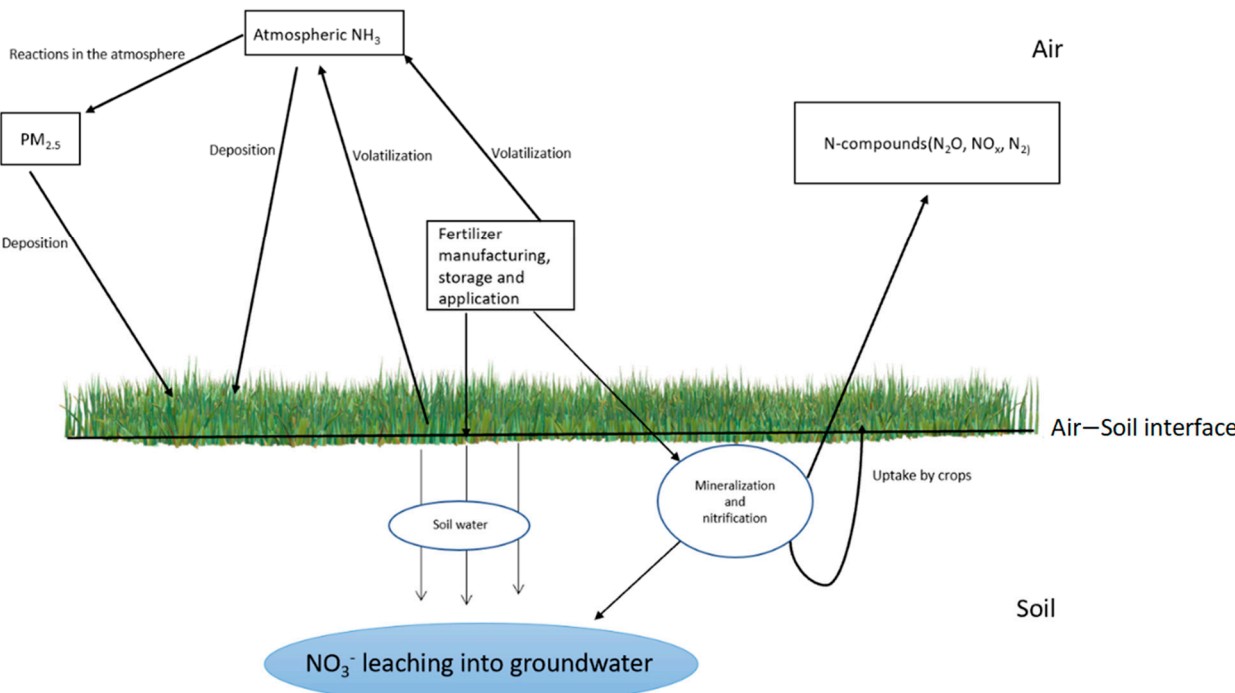

**Figure 1.** Abbreviated N cycle showing the role of $NH_3$ adapted from Doyle et al. [19].

The susceptibility of $NH_3$ to volatilize from fertilizer is largely driven by the alkalinity of the zone surrounding the granule or droplet of fertilizer as it reacts with the soil [39]. Other factors, such as nitrification, plant uptake, immobilization and exchange in the soil can also reduce volatilization potential [38]. $NH_3$ emissions near intensive arable agricul-

tural sources are known as "hot-spots"; however, generally, $NH_3$ concentrations reduce to background levels as the ammonia is dispersed on the surrounding landscape as well as emitted to the atmosphere where it undergoes atmospheric reactions and transport [40].

*2.2. Atmospheric Chemistry of $NH_3$ and SIA Formation*

Agricultural systems are inclined to concentrate N, with the use of either organic or synthetic fertilizers, with subsequent emission of $NH_3$ into the atmosphere. As $NH_3$ enters the atmosphere, it generally moves laterally with a relatively short half-life, and can be deposited within a small radius (a few hundred meters) of the source clinging to nearby surfaces [41]. However, the residence time of $NH_3$ is dependent on various factors, such as the conversion rate of $NH_3$ to $NH_4^+$ and the rate of deposition or decomposition of each species [42]. A residence time of between 0.8 and 4 days for $NH_3$ and between 5 and 19 days for $NH_4^+$ is generally accepted, after which they are deposited back to ground level [19,43].

Ambient atmospheric $NH_3$ can undergo deposition in three major forms: $NH_3$ gas is returned to the surface by dry deposition, it is deposited as an aerosol in submicron atmospheric water droplets forming a salt in association with other pollutants (this is not to be confused with SIA formation, where $NH_3$ undergoes a neutralization reaction with oxides of sulphur and nitrogen), and as $NH_4^+$ in the form of wet deposition [44]. Dry deposition refers to ambient $NH_3$ gas being directly deposited back to ground level [45]. This is owed to the translational kinetic energy of particles competing with gravitational forces. However, given the density of a gas such as $NH_3$ it decreases with increasing altitude in the atmosphere [46]. The main driving force for dry deposition of $NH_3$ gas from the atmosphere, therefore, is turbulent diffusion. This may be affected by near surface winds, atmospheric stability, surface roughness, density profile and spatial distribution of sources of $NH_3$ [47].

Notably, $NH_3$ flow is not unidirectional, but may flow in both directions (flux) between the atmosphere, vegetation and soils (both soils and vegetation may emit as well as absorb $NH_3$ gas) [48–50]. As shown in Figure 2, there are compensation points in the leaf stomata and the soil which have their own concentration levels. Thus, the fluxes through these pathways are bidirectional; depositing if the air concentration exceeds the surface compensation concentration, and emitting if the surface concentration is in exceedance [51–53]. The surface compensation concentration in soil and leaves is dependent on the $NH_3$ concentration in the soil pore (air space) or the stomatal cavity in leaves, being in equilibrium with aqueous $NH_4^+$ ions and hydrogen ($H^+$) ions in solution in the soil water or the apoplast leaf tissue, respectively [52–54]. Sutton et al. (1992) found $NH_3$ emission to be favoured during warm, dry conditions, and deposition to be favoured during cool, wet conditions [55]. This is due to the relationship between $NH_3$ on leaf surfaces and the presence of water on the cuticle [2]. A study by Sutton et al. (1993) also noted similarities in patterns of deposition to melting snow and wet vegetation as those found over unfertilized vegetation with canopy resistance less than 30 sm$^{-1}$ [56]. The reduced deposition velocities that occurred during some runs of this experiment were probably a consequence of the surface being frozen, although these might have resulted from an increase in either surface resistance or surface concentration.

Ambient atmospheric $NH_3$ can also rapidly transform into $NH_4^+$ due to reactions with water present in the atmosphere. Normally, there is less $NH_3$ present in the atmosphere compared with $NH_4^+$, except at localised hot spots, where large quantities are volatilized [57]. Wet deposition removes $NH_4^+$ from the atmosphere through two main processes, namely, nucleation scavenging and impact scavenging. Nucleation scavenging occurs when particles act as cloud condensation nuclei [58]. As water accumulates on the particle, the aerosol may increase in size until the plume (fog) droplets deposit on the Earth's surface or fall from the air as precipitation. When the plume is combined with a cloud of water droplets, the $NH_4^+$ can be relocated into these droplets [59]. These aggregate, by various microphysical processes, to form raindrops or even snowflakes and are deposited from the atmosphere [2]. This is a more efficient deposition pathway for $NH_3$; however, it

differs from the in-cloud scavenging of $NH_4^+$ aerosols (SIAs), as measurements of $NH_4^+$ wet deposition are needed to interpret wet deposition data for $NH_4^+$. The deposition also depends on an accurate description of wet scavenging (both in-cloud and below-cloud) [41]. This occurs by physical contact or, in the case of $NH_3$, through absorption due to its high solubility, with the much larger droplets of precipitation [58]. Impact scavenging is one of the atmosphere's cleansing processes, and this removal process determines the chemical composition of precipitation [45]. While many studies have focused on the relationship between concentrations of gas and/or particles in the atmosphere measured at the surface and the corresponding concertation of ions in precipitation collected [60–62], few studies have investigated the changes which occur in gas and particle concentrations in the air during separate precipitation events. Mountains can also heighten SIA concen-trations by trapping pollution that may alternatively be advected away from a given area [63,64]. Precipitation as well as cloud water are naturally acidic [65–67]; thus, most of the $NH_3$ scavenged by drops reacts with a hydrogen ion ($H^+$) to form $NH_4^+$ [57]. Precipitation can, henceforth, be considered as a potential component of 'acid rain', using the term in its broadest sense [41].

| | | | |
|---|---|---|---|
| ⌇ Resistance to deposition | | $r_a$ | Aerodynamic resistance |
| $\chi s$ | Stomatal compensation point | $r_b$ | Laminar boundary layer resistance |
| $\chi c$ | Canopy compensation point | $r_{ic}$ | In-canopy resistance |
| $\chi a$ | Atmospheric compensation point | $r_s$ | Stomatal resistance |
| $\chi soil$ | Soil compensation point | $r_{soil}$ | Soil resistance |
| | | $r_c$ | Cuticular resistance |
| | | $r_{glbl}$ | Ground laminar boundary layer resistance |

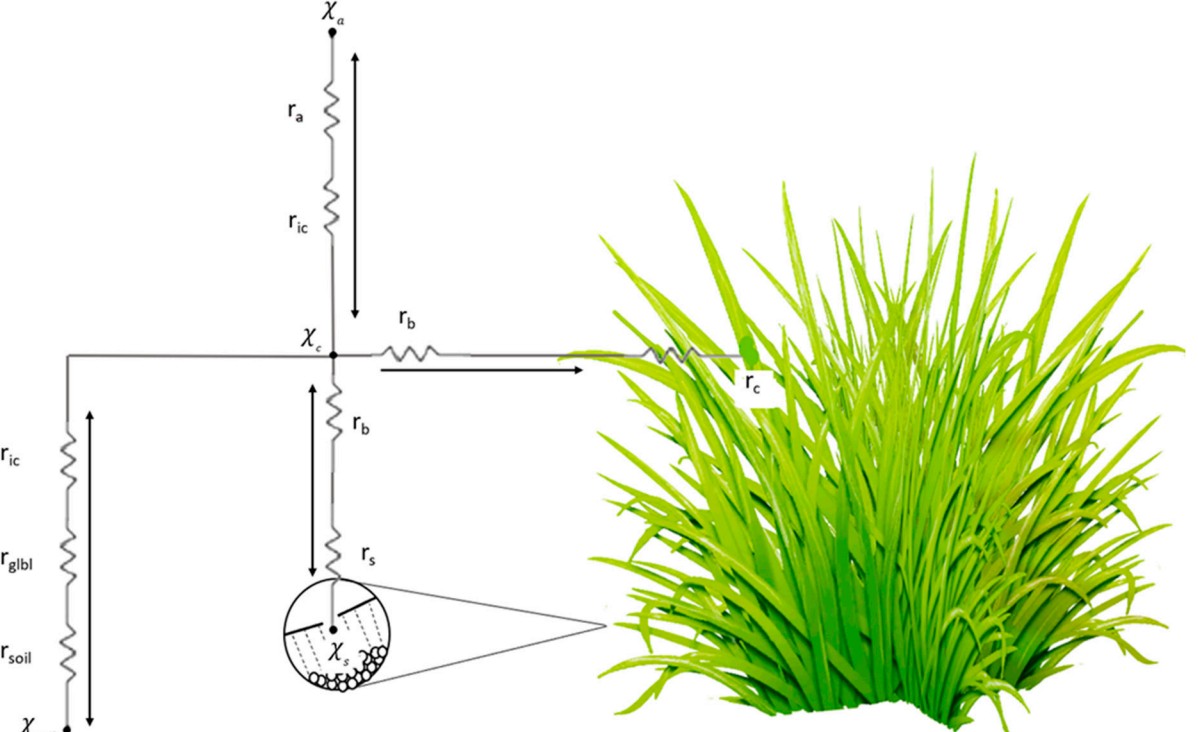

**Figure 2.** Resistance model schematic for bi-directional $NH_3$ flux adapted from Pleim et al. (2013) [52].

## 2.3. Atmospheric Chemistry of PM Formation

In the atmosphere, $NH_3$ gas can react with sulphur dioxide ($SO_2$) and $NO_x$ to form aerosols. $SO_2$ and $NO_x$ can also undergo oxidation in the atmosphere, forming sulfuric acid ($H_2SO_4$) and nitric acid ($HNO_3$) which are neutralized in the atmosphere by $NH_3$. These neutralization reactions result in ammonium sulphate (($NH_4)_2SO_4$), ammonium bisulphate (($NH_4)HSO_4$) and ammonium nitrate ($NH_4NO_3$). These salts are commonly referred to as SIAs [66–71]. Aerosols, also known as particulate matter (PM), are generally broken down into two main groups for monitoring purposes: a coarse fraction (particle size of 2.5–10 µm,) and a fine fraction (particle sizes between 2.5 and 0.1 µm). More recently, the terms $PM_{2.5}$ and $PM_{10}$ have been used and denote particles less the 2.5 and 10 micron in size, respectively. In Europe, secondary PM, including SIAs and secondary organic aerosols (SOAs), contribute to an estimated 70% of the background concentrations of particulate matter in the 2.5 µm size fraction alone [72]. $NH_3$ aerosols comprise a significant portion of SIAs present in the atmosphere, accounting for 30–50% of aerosol mass of $PM_{2.5}$ and $PM_{10}$ [2,40,72].

Atmospheric PM consists of inorganic and organic species such as sulfate, nitrate, chloride, water content, soil dust, elemental carbon, and organic carbon [73]. Primary sources of SIAs (emissions which do not undergo reactions in the atmosphere, but are directly emitted) include wildfires, geogenic (wind erosion) and biogenic sources [74]. Anthropogenic sources of primary SIA comprise industrial processes, combustion, electric utility (combustion sources), residential emissions (burning of coal and peat), construction (fugitive sources), and vehicular emissions [68,75–77].

Ambient atmospheric SIAs, such as $NH_3$ and $NH_4^+$, can be removed from the atmosphere through wet and dry deposition [78]. The dry deposition of aerosol $NH_4^+$ is somewhat different to that of gaseous $NH_3$ as atmospheric turbulence dominates the transport from the atmosphere to the laminar boundary layer. Mechanisms such as Brownian motion, inertial impaction, interception, phoretic forces, etc., also play key roles in the deposition of PM; however, they act differently depending on the size of PM (i.e., whether it is $PM_{10}$ or $PM_{2.5}$) [59,79,80]. The size of $PM_{2.5}$ also ensures that re-emission into the atmosphere does not occur easily; hence, PM flux, unlike $NH_3$ flux, is unidirectional [81]. Ambient atmospheric SIA wet deposition processes are collectively known as wet scavenging. Wet scavenging is an essential process for the maintenance of balance between sources and sinks of SIA [82]. Wet scavenging of atmospheric particulate matter (PM) occurs through two notable processes: below-cloud scavenging (washout) and in-cloud scavenging (rainout) [83].

In the process of wet deposition, particles are incorporated into hydrometeors before being brought back to the surface in aqueous form [84]. Similarly to dry deposition, both in-cloud and below-scavenging is highly dependent on the size of particulate matter, with rates of removal differing for each size fraction [2,85]. Consequently, coarse particles are deposited near source areas while fine and ultrafine fractions are transported away from sources prior to deposition [84,86,87]. Monitoring at both national and EMEP scale indicates that Ireland has a number of important transboundary pollution pathways, namely, from the United Kingdom, mainland Europe and North America, although pollution sources also arise from Africa, especially during springtime when elevated levels of Saharan dust can be detected in Ireland [88]. Atmospheric PM can also serve as cloud condensation nuclei (CCN) [89–96]. The condensation of nitric acid on aerosol particles may enhance aerosol activation to cloud droplets by providing additional soluble material to the particle surface, as well as elevat-ing the water uptake and growth of aerosol particles [97–104]. Under favourable meteorological conditions, hygroscopic water molecules are attracted to the particles present in the atmosphere, leading to a rapid increase in mass fraction [105]. The process of the hygroscopic growth process can be described by Köhler's theory of water vapour condensation, forming liquid cloud drops based on equilibrium thermodynamics [106]. This process plays a key role in cloud physics, making atmospheric PM a

vital element in understanding cloud formation, as well as the role it plays in the Earth's climate systems.

*2.4. Controlling Factors of Emission and Transport*

Emissions for pollutants are controlled by various factors. Biotic factors represent all living things which affect emissions, such as the flora and fauna of environments. Abiotic factors refer to all the non-living factors which can affects emissions, for example meteorology and climate. Anthropogenic activities also affect the emission of pollutants through their interaction with controlling factors. It must be acknowledged that, while biotic and abiotic factors play major roles in emission control of SIA, the biggest control factor for the formation of secondary pollutants will be the availability of precursor gases. Therefore, it can be stipulated that anything which may control and/or affect $NH_3$ emissions, will have an indirect effect on SIA formation.

Notably, anthropogenic activities such as agriculture have the ability to influence and even alter the N cycle between these three major pools by additional N loading through the use of synthetic fertilizers to soils and vegetation. The N cycling of these additions is affected by climate, soil properties to which the fertilizer is added, vegetation type covering the soil and the management of agricultural activities [107]. Soil texture, pH, carbon to nitrogen (C:N) ratio, soil organic matter (SOC) content and moisture content exhibit significant control on the soil's ability to cycle various forms of N.

Climate (on both local and global scales) and pollutant emissions have a cyclic effect on one another. Global and local climate are affected by the emission of pollutants and the rate of emissions are affected by local meteorology and climate. The terrestrial N cycle has been drastically modified by global climate change as a result of increasing agricultural intensification and fossil fuels [4]. Microbial processes involved in denitrification and nitrification are affected by climate change at a local and global level, resulting in serious environmental issues, such as elevated $NO_3^-$ leaching and NOx emissions [108].

Similarly, the formation and transport of SIA is also affected by local meteorology and climate change. Ambient relative humidity (RH) and temperature are key meteorological factors for the determination of the state of SIA. For example, reactions of $NH_3$ with $HNO_3$ at low tem-peratures will show a shift in the equilibrium of the system towards the aerosol phase. Low RH results in $NH_4NO_3$ to form as a solid particle [109,110]. As temperature increases, the air's capacity to hold moisture also increases, resulting in RH decreasing [111]. These changes can affect SIA dynamics such as transport pathways, and formation on a localised basis [112,113]. The specifics of the sources and causes of locally high levels of PM are singular to each location. In Philadelphia, PA, Cheng et al. (1992) found that high pressure associated with maritime topical and non-polar continental air masses generated the highest total particulate matter concentration. These air masses were defined by high pressure and temperature values, high dew points, percent of clear sky and stability [46]. Usually, high-pressure systems develop after the passage of a cyclonic system [114]. Sporadically, low-pressure systems can develop as opposed to high-pressure systems with the passage of a storm, resulting in high PM levels if the winds associated with the storm stir up dust and/or other particulates [114–116]. Lower PM concentrations are generally correlated with polar and moderate air masses. Depending on weather conditions (precipitation, wind direction, wind speed, etc.), these air masses occur ahead of a low-pressure system [117]. These air masses are generally advected from the Atlantic Ocean in Ireland. Terrain also affects the specific weather patterns which influence SIA levels. Mountains and canyons can increase atmospheric stability, and thus increase SIA concentrations in the neighbouring valleys due to cold air drainage [118]. Augmented stability in valleys is most prevalent under synoptic high-pressure conditions [119].

## 3. Linking the Soil–Water–Atmosphere Nexus

The development, parameterization and validation of $NH_3$ models over the years, has been based on steadily emerging data for $NH_3$ concentrations in a broad range of

ecosystems and the atmosphere and the associated flux values across all scales [120]. At sub-ecosystem scales (chamber, plot, field), this has stemmed from technological advances in $NH_3$ measurement and analysis, both quantitatively and qualitatively.

The use of flux measurement instrumentation capable of lower detection limits than was available before, while also selectively quantifying gaseous $NH_3$ from aerosol $NH_4^+$, enables more accurate measurements [121–129]. This is particularly valid on a field scale, using Bowen ratio techniques at remote background locations (i.e., where sub-parts per billion levels of $NH_3$ are present) and for over-fertilized agricultural ecosystems, which has helped generate many exchange datasets [130,131].

The key mechanisms and controls of $NH_3$ exchange have been determined at substrate, plant, and ecosystem level, although a substantial gap in knowledge remains regarding the complete $NH_3$ cycle. This can partially be attributed to the lack of regulation of $NH_3$ as a gaseous atmospheric pollutant. Compared to other atmospheric gaseous pollutants such $SO_2$, $NO_x$ and volatile organic compounds (VOCs), no extensive control measures have been put in place for the control and mitigation of $NH_3$ emissions [2]. Indeed, there are currently very few regulations in place, and incentive programmes to reduce emissions are highly lacking in many countries globally, including Ireland. This is all despite the contribution $NH_3$ makes to the overall atmospheric particulate matter mass loading. In fact, Ireland has been implementing policies which are contrary to the reduction and mitigation strategies which should be in place, with schemes such as the Food Harvest 2020 [132] and Food Wise 2025 [132] which boast agricultural intensification. This has resulted in atmospheric $NH_3$ concentrations exceeding the permitted levels from 2016 over the subsequent five years.

### 3.1. Direct Source Measurement: State-of-the-Art Techniques Currently in Use

Measurement techniques of atmospheric $NH_3$ have improved over the last two decades. One major difficulty when developing measurement techniques for atmospheric $NH_3$ arises from the simultaneous presence of $NH_3$ gas and $NH_4^+$ in the form of PM (liquid and solid state) [123]. Additionally, variations in ambient atmospheric $NH_3$ concentrations and the ability of $NH_3$ gas to interact with surfaces [123,125] present further difficulties when developing techniques for the measurement of atmospheric $NH_3$.

The most widely used techniques for $NH_3$ measurement are denuder sampling techniques and diffusive samplers. A definition given by Doyle et al. (2013) describes diffusive samplers as devices which are capable of taking samples of gas or vapor pollutants from the atmosphere at a rate controlled by a physical process, such as diffusion through a static air layer or permeation through a membrane of the air through the sampler [19]. Diffusive sampling relies on the mass flux of substances from regions of high concentration to regions of low concentration. Denuder sampling techniques such as the Annual Denuder Method (ADM) have proven to be successful for $NH_3$ gas sampling. Denuders work based on a laminar airstream passing through a suitably long tubular enclosure whose walls are coated in the appropriate sorbent for a given acidic or basic gas present in the atmosphere [133]. The sampler also has a capacity to differentiate between $NH_3$ as part of SIA and $NH_3$ gas [134,135]. Despite widespread use, both denuder techniques and diffusive sampling come with limitations such as relatively low time resolution, labour-intensive sampling procedures and post-sampling wet chemistry analysis being required, which can introduce contaminants to the samples during periods of storage and/or analysis [123]. Despite these limitations, denuders and diffusive samplers remain the most cost-efficient sampling techniques for atmospheric sampling of $NH_3$ and SIA.

Other methods for measuring ambient atmospheric $NH_3$ are spectroscopic techniques such as photoacoustic spectroscopy (PAS) [136–138], differential optical absorption spectroscopy (DOAS) [126,129], tuneable diode laser absorption spectroscopy (TD-LAS) [129,139] and chemical ionization mass spectroscopy (CIMS) [121]. These techniques rely on infrared (IR) or laser-based detection such as laser diode detectors which can single out $NH_3$ gas. Differences in accuracy and precision of the instrumentation used for the

measurement of ambient atmospheric $NH_3$ arise from differences in inlet length, the calibration frequency of each instrument and the frequency of changes in collection vessels such as filters or diffusion tubes [122].

A comparison of diffusive samplers and denuders performed by Sutton et al. (2001) found that passive diffusion tubes were imprecise for the measurement of $NH_3$, several more complex methods for sampling ammonia are available, including automatic batch denuders, continuous denuders and diffusion scrubbers. However, each of these are highly expensive and would be inappropriate for monthly sampling at many sites [140]. They also found the most precise method to be sampling using the denuder technique with a time resolution of two weeks and ambient concentrations of >2 μg m$^{-3}$ $NH_3$.

Another comparative study conducted by Von Bobrutzki et al. (2010) explored an array of techniques from spectroscopic to wet chemistry methods [125]. While differences were found in the concentrations measured, an overall high correlation of $R^2$ > 0.84 was found compared with the average of all instruments used. Correlation worsens when concentrations <10 ppb $NH_3$, due to differences in inlet length of samplers and time–response.

The Environmental Protection Agency (EPA) in Ireland currently monitors atmospheric $NH_4^+$ at three representative sites (Carnsore, County Wexford; Oak Park, County Carlow and Malin, County Donegal) in agreement with the European Monitoring and Evaluation Programme (EMEP); however, there is currently no continuous monitoring network in place for ambient atmospheric $NH_3$ gas concentration [141].

The UK National Ammonia Monitoring Network (NAMN), Northern Ireland, has three continuous monitoring sites for $NH_3$ gas in the atmosphere. $NH_3$ is also a pollutant which is currently not covered by the CAFE Directive under ambient air quality [20] and does not fall under the national ambient air quality network, which is managed by the EPA under a policy-driven programme for the Convention on Long-range Transboundary Air Pollution (CLRTAP) for international co-operation to solve transboundary air pollution problems within the EMEP [19].

### 3.2. Modelling of $NH_3$ and SIA

Various numerical models have been generated for the implementation of modified gradient techniques to infer the surface flux of $NH_3$ (and chemically reactive species) from field measurements, while also accounting for gas-to-particle interconversion (GPIC) and its effects on vertical flux divergence [52,53,142–148]. Modelling results presented in literature showed that atmospheric reactions could theoretically change $NH_3$ fluxes as much as 40% [128] or even cause flux reversal [142].

While most emission model studies focus on the influence of precursor species (e.g., $NH_3$) on aerosol concentrations; a study by Zöll et al. (2016) has shown a novel approach to $NH_3$ emission modelling, with an overall aim of creating a better understanding of geological and temporal aspects of emissions [144]. Models focusing on the prior, result in the models being too simplistic, with inaccuracies in estimating emissions, by not accounting for environmental factors affecting emissions overall. More precise model constructs can be achieved to include the evaporation process as a mechanism of action to improve performance where models with a forecasting element are the focus. This improved understanding has allowed for a more complicated model construct having higher accuracy and precision than other models of this type.

Many localised field experiments based around $NH_3$ deposition measure concentrations decreasing as a function of distance from the source. Dry deposition processes control the transfer of pollutants from the atmosphere to the surface [46]. Studies conducted in Denmark have made critical improvements in atmospheric models of $NH_3$, specifically in the development of a regional N deposition assessment model. The model was built by replacing static seasonal variations with dynamic applications accounting for physical processes (e.g., volatilization) and agricultural management practices such as seasonal timing of fertilization [149]. However, the data required for such a model to be constructed are insufficient in most European countries, as such inventories are poorly managed or

do not exist on a nation-wide scale. The state of the art of $NH_3$ surface–atmosphere exchange, in terms of measurement and modelling, has been investigated in a number for reviews [150–152]. Existing models of surface exchange are reviewed at different scales from leaf to the global level, with a focus on the development of canopy-scale models and their application at larger scales (regional). A large number of models have been generated to simulate $NH_3$ exchange fluxes for various ecosystem components (soil, leaf, plant, plant canopy, litter) or processes (heterogenous phase chemical reactions) [52,123,153,154]. These system dynamics have been modelled either individually or at a canopy-scale soil–vegetation–atmosphere basis. Larger scale (landscape, regional or global) models are 2D or even 3D, usually including simplified versions of canopy-scale models simulating 1D surface exchange, as part of the wider context including chemistry, emissions, dispersion, and deposition [53,123,155].

Canopy-scale models incorporate individual component processes and the interactions they undergo within the soil-vegetation-atmosphere framework [123,155–159]. The objective of this type of modelling is to determine the net ecosystem $NH_3$ flux from the following inputs: (i) ambient $NH_3$ levels; (ii) meteorology, or alternatively micrometeorological factors; (iii) ecosystem characteristics such as canopy height and leaf area index (LAI) [123,159]) There are many ways in which models have been developed to address system dynamics, some more mechanistic than others.

Several experimental campaigns have been carried in order to supply data for the dry deposition velocities for different types of pollutants (e.g., SIA) and deposition surfaces [81,160–165]. However, due to the issues associated with influencing factors which play a part in deposition velocity, differences between the data lead to a difficulty in generalising this phenomenon [81]. Due to this controversy, the dry deposition process cannot be studied using a single modelling approach. Indeed, the models proposed in the literature have limited ability of representing dry deposition phenomena as a whole for several categories of pollutants such as SIAs and its deposition surfaces (plant canopy, water surfaces, etc.) as their applications are only valid for a definitive set of conditions (certain types of climates, meteorology, topography, etc.) [81,161].

Most models are based on the resistance analogy, in which the flux ($F_x$) between two potentials, A and B, is equal to the potential difference divided by the resistance, with the atmosphere–soil being represented as a network of potentials connected by resistances in series for different layers and in parallel for different pathways [81]. Kinetics for the chemical source and/or sink associated with the $NH_3$–$HNO_3$–$NH_4NO_3$ triad are described either using chemical timescales, reaction rate coefficients or as a full model of size-resolved chemistry with the addition of microphysics [166,167]. The model developed by van Oss et al. (1998) described the above-mentioned reaction's shift towards equilibrium as a relaxation-type equation for the flux divergence. Atmospheric forecasting of pollution events is a recent development with a large research focus involving research institutes globally in model development [168]. While these types of models are still in the development stage, some of the first systems have appeared as "operational" systems. However, due to large disagreements between parameterization, these models are largely experimental and until a unified set of parameters are established on an EU-wide scale, these models will remain so.

The difficulties of unified parameters for modelling in the EU mainly arise from the differences of environmental factors between countries such as temperature effects of local meteorology, climate and geographical features which all affect emission and deposition processes. As a result, temperature effect is not taken into account in current European models according to Menut and Bessagne's (2010) review on Chemistry Transport Models (CTMs) [168]. A study by Skjøth et al. (2013) found that this is also the case for Chemistry Climate Models (CCMs) [40]. These studies are in agreement with the proposed theory of Undine Zöll et al. (2016) of improving models by applying the dynamic processes which result in spatio-temporal variations in emissions of pollutants such as $NH_3$ [153].

Most EU models receive data from the EMEP system, which is a gridded emission inventory. These inventories are constructed based on national emission factors integrated with gridded activity data such as number of animals on a national basis [40]. However, the data represented by the EMEP campaign are too generalised for accurate and precise model constructs to be achieved, especially for models with a forecasting element of total nitrogen load. This can currently be seen, as of the 27 air pollution prediction models in use in the EU with a temporal profile element used for the forecasting of $NH_3$ pollution, none have sufficient accuracy or precision, but in fact most either over or under-estimate ambient atmospheric concentrations [149].

As there is currently no continuous monitoring of ambient $NH_3$ concentrations in Ireland, most European-scale models exclude Ireland as there are insufficient data supplied for inclusion. In order for Ireland to be included in modelling campaigns, highly detailed data are required. A monitoring campaign based on the dynamics of $NH_3$ emission and deposition processes can provide such data, as well as providing clarification of SIA dynamics and transportation.

## 4. Concluding Remarks

Ambient atmospheric $NH_3$ is an important pollutant contributing significantly to SIA generation, a contribution which is often under-estimated due to the short-range transport of $NH_3$ from hot-spots and its short half-life in the atmosphere. Studies focusing on $NH_3$ measurement are often based on distance, meaning the distance $NH_3$ is transported from an area of interest, especially in deposition study models. The dynamics of $NH_3$ through the environment are poorly defined; thus, source and receptor interactions and effects are crudely understood at best. While a cause-and-effect relationship has been established between $NH_3$ and SIAs, mitigation strategies have started to recognise that the reduction in precursor gases inevitably serves the reduction in SIAs. There is a lack of understanding of system dynamics the precise nature of how to efficiently mitigate both species of pollutants is still not clearly understood.

Accurate data on the spatial and temporal distribution of $NH_3$ and SIA emissions are crucial input to models of atmospheric transport and deposition. This is particularly important when the resulting deposition maps are utilised to establish suitable mitigation strategies regarding ecosystem decline as a result of pollution. The accuracy and precision of prediction models is dependent on the quality of the input data; hence, there is a need for high-quality emission and deposition inventories for both species of pollutants. This would require analysis and monitoring of the dynamics of both pollutants rather than studies based around only precursor gases from which concentrations of secondary pollutants are extrapolated. However, long-term campaigns of direct measurements on which these inventories and models are based remain difficult to conduct.

All techniques of measurement are affected by the built-in bias of the design chosen whether spectroscopic or techniques based on wet chemistry methods. To these potential errors, additional errors arise from geographical features of the terrain where measurements are carried out as well as local meteorological conditions which can affect measurement. Choosing the most suitable method of measurement can minimise these errors, improving data quality by precise and accurate measurements, making it a crucial step in any research study. Therefore, all studies based around $NH_3$ pollution and SIAs arising from $NH_3$ should take these factors into consideration. Furthermore, studies with the aim of contributing towards a continuous monitoring system for Ireland have to consider the quality of data that the monitoring network would provide, as it would have to be sufficient to contribute not only at a national, but at a European-wide scale.

**Author Contributions:** Conceptualization, V.P. and D.J.O.; methodology, A.G., E.M., E.J.M., J.C. and V.B.; formal analysis, V.P. and D.J.O.; investigation, V.P. and F.N.; resources, V.P.; data curation, V.P.; writing—original draft preparation, V.P.; writing—review and editing, F.N.; visualization, V.P. and A.G.; supervision, D.J.O. and S.H.; project administration, D.J.O. and S.H.; funding acquisition, A.G. All authors have read and agreed to the published version of the manuscript.

**Funding:** This research was funded by EPA Research and the Department of Agriculture, Food and Marine (DAFM), grant number 2021-HE-1052.

**Institutional Review Board Statement:** Not applicable.

**Informed Consent Statement:** Not applicable.

**Data Availability Statement:** Not applicable.

**Conflicts of Interest:** The authors declare no conflict of interest. The funders had no role in the design of the study; in the collection, analyses, or interpretation of data; in the writing of the manuscript; or in the decision to publish the results.

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
