# Peer review of "Ammonia Cycling and Emerging Secondary Aerosols from Arable Agriculture: A European and Irish Perspective"

_2813-4168, doi:10.3390/air1010003_

Round 1
Reviewer 1 Report
This manuscript is well written and worthy of publication. This reviewer has two minor comments for the authors considering: 1) all modeling works on NH3 air/soil exchange don't include the effect of deposited NH3 on changing soil NH3 emission potentials;this would affect the model performance on NH3 mixing ratios; 2)Increased mixing ratios of atmospheric NH3 over the intensive agriculture zone sometimes play a role in suppressing evaporation of ammonium nitrate instead of its formation. In fact, high concentrations of ammonium nitrate can be rapidly formed in cooling combustion plumes within dozens of seconds and few minutes when they just leave the stack (Shen et al., 2022, Atmosphere, 13 (5), 673). At least to this reviewer, the highly concentrated ammonium nitrate would firstly experience evaporation process prior to its secondary ambient formation.
Reviewer 2 Report
What is the novelty of this study?
It seemed that the control and experiment are clearly missing in this article.
The citation is not in the correct format for the reference list.
The section numbers do not seem to be within the journal format.
Line 37 – 1. Introduction
Line 70 – 2. Precursor species’ dynamics and SIA formation
Line 307 – 3. Linking the soil-water-atmosphere nexus
Line 471 – 4. Concluding Remarks
Please double-check the journal requirement before submitting it for publication.
Round 2
Reviewer 2 Report
Although the authors did not make any effort to improve the manuscript, I have no objection for the manuscript to be published.
Author Response
A paragraph has been added from line 480 to include some review on thermodynamic models about NH3-ammonium nitrate partitioning.